# Coupled Recurrent Models for Polyphonic Music Composition

## Abstract

This work describes a novel recurrent model for music composition, which accounts for the rich statistical structure of polyphonic music. There are many ways to factor the probability distribution over musical scores; we consider the merits of various approaches and propose a new factorization that decomposes a score into a collection of concurrent, coupled time series: "parts." The model we propose borrows ideas from both convolutional neural models and recurrent neural models; we argue that these ideas are natural for capturing music's pitch invariances, temporal structure, and polyphony.

We train generative models for homophonic and polyphonic composition on the KernScores dataset (Sapp, 2005), a collection of 2,300 musical scores comprised of around 2.8 million notes spanning time from the Renaissance to the early 20th century. While evaluation of generative models is known to be hard (Theis et al., 2016), we present careful quantitative results using a unit-adjusted cross entropy metric that is independent of how we factor the distribution over scores. We also present qualitative results using a blind discrimination test.

## 1 Introduction

The composition of music using statistical models has been strongly influenced by developments in deep learning; see Briot et al. (2017) for a recent survey of this field. Previous work in this field mostly focuses on either monophonic scores (Sturm et al., 2016; Jaques et al., 2017; Roberts et al., 2018), rhythmically simple polyphonic scores (Liang et al., 2017; Hadjeres et al., 2017; Huang et al., 2017), or rhythmically simplified encodings of more complex scores (Boulanger-Lewandowski et al., 2012). In this paper we consider rhythmically complex, polyphonic scores. To this end, we propose a new generative modeling task on the KernScores dataset, a diverse collection small-ensemble western music. We seek to understand *how well we can model the local distribution of rhythmically complex, polyphonic music*.

Many notable successes in deep learning are achieved in domains where natural weight-sharing schemes allow models to borrow strength from similar patterns in different locations: convolutions in vision, for example, or autoregressive models in language. Polyphonic music has rich spatial and temporal structure that is potentially amenable to such weight-sharing schemes. In this paper, we will consider how we should factor the probability distribution over scores to allow our models to take advantage of shared patterns, and also *how to construct models that explicitly leverage these patterns with shared weights*.

In Section 2 of this paper, we will discuss previous approaches to monophonic and polyphonic music composition. In Section 3 we introduce two new generative modeling tasks on the KernScores dataset: a single-part, homophonic prediction task and a multi-part, polyphonic prediction test. We additionally discuss quantitative evaluation and introduce an adjusted cross-entropy rate that is invariant to the approach we use to factor the score. In Section 4 we discuss several approaches to factoring the distribution over scores, and propose a new approach that exploits the structure of music to allow us to share weights in our models. Section 5 proposes models for this factored distribution and identifies opportunities for weight-sharing. We present quantitative and qualitative evaluations of these models in section 6.

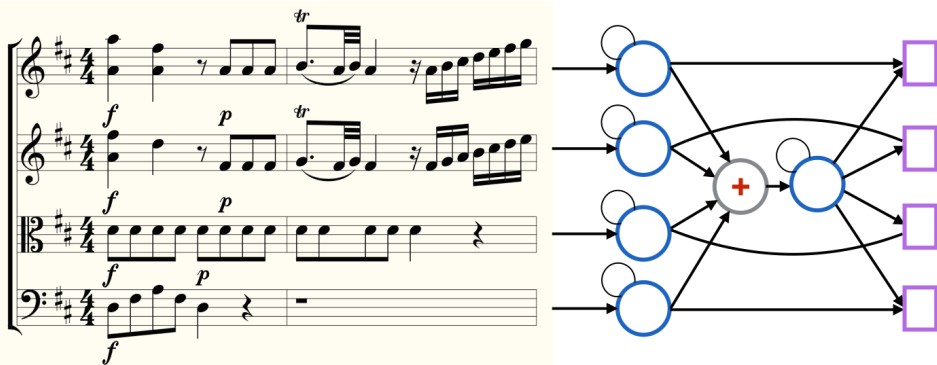

Figure 1: Coupled state estimation of Mozart's string quartet number 2 in D Major, K155, movement 1, from measure 1, rendered by the Verovio Humdrum Viewer. A representation (blue) of the state of each part is built at each time step, based on the previous state's representation and the current content of the part. A representation of the global state of the score is built from the previous global state and a sum (red) of the current states of each part. For each part, new notes (purple) are predicted using features of the global representation and the representation of the relevant part.

## 2 RELATED WORKS

Early efforts to build statistical models of music focus on single part, monophonic sequences (melodies). Possibly the first statistical model for music generation was proposed by Pinkerton (1956). This work was followed concretely by Brooks et al. (1956), who built a Markov transition model estimated on small music corpora. A proliferation of work on computer-generated music and data-driven musicology followed these pioneering works in the 1960's and 1970's; see Roads (1980) for a survey. An important development during this era was the application of Chomsky-inspired grammatical analysis to music, exemplified by Kohonen (1989); this latter work contemplates the generation of two concurrent musical parts, one of the earliest examples of polyphonic generation.

The first application of neural networks to algorithmic melody composition was proposed by Todd (1989). This work prompted followup by Mozer (1994), who altered the representation of the input to Todd's model using pitch geometry ideas inspired by Shepard (1982); the relative pitch and note-embedding schemes considered in the present paper can be seen as a data-driven approach to capturing some of these geometric concepts. Neural melody generation was revisited by Eck & Schmidhuber (2002), using long short-term memory models. More recent work on melodic composition experiments with techniques to capturing longer-term structure than classic recurrent models provide. Jaques et al. (2017) explore reinforcement learning as a tool for eliciting long-term structure, expanding on ideas first considered by Franklin (2001). Roberts et al. (2018) also attempt to capture long-term structure, proposing a variational auto-encoder for this purpose.

The work on polyphonic music is considerably younger. The aforementioned work of Kohonen (1989) considers two-part composition. Another early precursor to polyphonic models was introduced by Ebcioğlu (1988), who proposed an expert system to harmonize 4-part Bach chorales. The harmonization task became popular, along with the Bach chorales dataset. See Allan & Williams (2006) for a classic discussion of this problem. Lavrenko & Pickens (2003) directly address multi-part polyphony, albeit using a simplified preprocessed encoding of scores that throws away duration information.

Maybe the first work with a fair claim to consider polyphonic music in full generality is Boulanger-Lewandowski et al. (2012). This paper proposed a coarse discrete temporal factorization of musical scores (for a discussion of this and other factorizations, see Section 4) and explored a variety of neural architectures several music datasets (including the Bach chorales). Many works on polyphonic models since 2012 have focused on the dataset and encoding introduced in Boulanger-Lewandowski et al. (2012), notably Vohra et al. (2015) and Johnson (2017). These works typically focus on quantitative log-likelihood improvements, and the degree to which these quantitative improvements correlate with quality is less clear.

For the Bach dataset, qualitative success is more definitive. Recently, concurrent work by Liang et al. (2017) and Hadjeres et al. (2017) proposed models of the Bach chorales with large-sample discrimination tests demonstrating the convincing quality of their results. In this line of work, quantitative results are lacking; Liang et al. (2017) learn a generative model and could in principle report cross entropies, although their work focuses on the qualitative study. The system proposed in Hadjeres et al. (2017) optimizes a pseudo-likelihood, so its losses cannot be easily compared to generative models. Quantitative metrics are reintroduced for the chorales in Huang et al. (2017). Both the latter papers propose non-sequential Gibbs-sampling schemes for generation, in contrast to the ancestral sampler used by Liang et al. (2017). Huang et al. (2017) make the case that a non-sequential sampling scheme is important for generating plausible compositions.

## 3 DATASET AND EVALUATION

The models presented in this paper are trained on data from the KernScores library (Sapp, 2005), a collection of early modern, classical, and romantic era digital scores assembled by musicologists and researchers associated with Stanford's CCARH.[1] This dataset consists of over 2,300 scores containing approximately 2.8 million note labels. Tables 1 and 2 give a sense of the contents of the dataset.

| Bach | Beethoven | Chopin | Scarlatti | Early | Joplin | Mozart | Hummel | Haydn |
|---|---|---|---|---|---|---|---|---|
| 191,374 | 476,989 | 57,096 | 58,222 | 1,325,660 | 43,707 | 269,513 | 3,389 | 392,998 |

Table 1: Notes in the KernScores dataset, partitioned by composer. The "Early" collection consists of Renaissance vocal music; a plurality of this collection is composed by Josquin.

We contrast this dataset and its Humdrum encoding with the MIDI encoded datasets used by most of the works we have discussed in this paper (a notable exception is Lavrenko & Pickens (2003), who used data derived from the same KernScores collection consided here). MIDI is an operational format in the sense that it consists of a stream of instructions that describe a musical performance. Indeed it was designed as a protocol for communicating digital performances, rather than digital scores. For example, it cannot explicitly represent concepts such as quarter-notes or eighth-notes or rests, only notes of a certain duration or the absense of notes. Heuristics are necessary to display a MIDI file as a visual score and, if the MIDI wasn't prepared specifically for this purpose, these heuristics are liable to fail badly. For example, a MIDI file might encode "stacatto" articulations by shortening the length of certain notes; the heuristics to determine the value of a note (quarter-note, eighth-note, etc.) based on its length become exceedingly complicated in such cases.

| Vocal | String Quartet | Piano |
|---|---|---|
| 1,412,552 | 820,152 | 586,244 |

Table 2: Notes in the KernScores dataset, partitioned by ensemble type.

It is not impossible to construct a high-quality dataset of MIDI scores; but the Humdrum format, designed consciously by musicologists to encode scores, helps to ensure the quality of the data by enforcing constraints that are absent from MIDI. Polyphonic music needs a new benchmark dataset. As Huang et al. (2017) point out, the dataset introduced by Boulanger-Lewandowski et al. (2012) is too coarsely preprocessed to continue to serve this purpose. And the Bach chorales dataset is too small to sustain much further research. The KernScores collection considered here is readily available, reasonably large, and is structurally guaranteed have high quality.

### 3.1 EVALUATION

Let $p$ denote the unknown distribution over musical scores $S$, and let $q$ be our model of $p$. We want to measure the quality of $q$ by its cross-entropy to $p$. Because the entropy of a score grows with its

---

[1] http://kern.ccarh.org/

length $T$, we will consider a cross-entropy rate.[2] By convention, we measure time in units of beats, so our cross-entropy rate will have units of bits per beat. Defining cross-entropy for a continuous process generally requires some care. But in the case of music, we observe that notation occurs at rational points in time, and for rational durations. We can therefore quantize time using the finest denominator $\Delta$ of times that appear in the dataset and define

$$H(p||q) \equiv \mathop{\mathbb{E}}_{S \sim p} \left[ -\frac{1}{T\Delta} \log q(S_0, S_\Delta, \dots, S_{T\Delta}) \right]. \tag{1}$$

This quantity is invariant to further refinement of the discretization. Suppose we quantize $S$ at a rate $\Delta/2$; then

$$q(S_0, S_{\Delta/2}, \dots, S_{2T\Delta/2}) = \sum_{k=0}^{2T} \log q(S_k | S_0, \dots, S_{(k-1)\Delta/2})$$

$$= \sum_{k \text{ even}}^{2T} \log q(S_k | S_0, \dots, S_{(k-1)\Delta/2}) = q(S_0, S_\Delta, \dots, S_{T\Delta}).$$

The odd terms vanish under our assumption that $\Delta$ was the finest denominator of notation in the dataset. We can think of $\Delta$ as the resolution of the score process.

Observe that Definition 1 is independent of any choice about how we factor $p$: it is a cross entropy measure of the joint distribution. As we will discuss in Section 4, there are many ways to construct a generative model of scores. These choices lend themselves to different natural cross-entropies with their own units, depending on how we factor. By converting to units of bits per beat given by Definition 1, we can compare results under different factorizations.

## 4 FACTORING THE DISTRIBUTION OVER SCORES

Polyphonic scores consist of features (notes and other notation) of variable length that overlap each other in quasi-continuous time. To factor the probability distribution over scores, we must somehow impose a sequential order upon the data. There is a loose partial order on scores implied by time but, in contrast to language, this order is not total. This slack admits many reasonable ways to factor the distribution over scores.

Most previous work factors a score by discretizing time. As we discussed in the previous section, there is a time resolution to all musical processes and we can discretize at this resolution without losing information. From this perspective, music looks like a large binary matrix of notes crossed with time; entry $(t, n)$ in this matrix indicates whether note $n$ is on at time $t$.[3] We can then generate music one slice of time at a time, generating a slice of time with a vector-valued prediction as in Boulanger-Lewandowski et al. (2012) or imposing an order (e.g. low to high) on notes and further factoring into binary predictions as in Liang et al. (2017).

The above factorization, while popular, is computationally difficult for rhythmically complex music. The process resolution $\Delta$ required to discretize scores without loss of information is the common denominator of all rhythmic events in the corpus. A corpus that contains both triplets and thirty-second notes, for example, would require a discretization of 48 positions per a beat. The datasets considered in Boulanger-Lewandowski et al. (2012) are discretized at either 1 or 2 positions per beat; as discussed by Huang et al. (2017), this downsampling is quite destructive to the structure of music, more analogous to dropping words from a sentence than downsampling an image. Hardware capabilities may eventually overcome the computational obstruction to discretized score modeling, but we take a different approach in this paper that scales better with rhythmic complexity.

One alternative to discretizing time and predicting notes at each time step is instead to operationalize scores. From this perspective, a score becomes a long sequence of instructions: start playing C, start playing E, advance to the next time step, stop playing C, etc. We can think of this approach as a run-length encoding of the discrete factorization. It has been considered recently for the related task

---

[2]This is analogous to the adjustment of cross-entropy by sentence length used in language modeling.

[3]We actually need two bits at each $(t, n)$ entry to distinguish between repeated notes and held notes: see Appendix B, Figure 7.

of modeling expressive performances by Oore et al. (2018). A similar factorization was proposed by Walder (2016), although that work does not implement a complete model. Operationalized run-length encodings greatly reduce the computational costs of learning and generation, at the expense of segmenting a score into highly non-linear segments. The number of items in the sequence between the beginning and the end of a note depends on how many other notes begin or end in the interim. Contrast this with the discrete factorization, for which every quarter-note (for example) lasts for exactly $1/\Delta$ time slices.

In this paper, we adopt a factorization inspired by the operationalized perspectives. First, we decompose a polyphonic score into a collection of parts. We can loosely think of a part as the set of notes in a score assigned to a particular voice or instrument.[4] Each part is homophonic and therefore run-lengths in a part correspond to the duration of notes (in contrast to operationalized full scores, where run-lengths have no musical interpretation). We learn to predict the next homophony (note, chord, or rest) in a part, making a prediction in the part that has advanced the least far in time, and breaking ties between parts by an arbitrary (fixed) order that we impose on the parts in each score. To predict the homophony, we impose an order pitches from low to high frequency, and make a sequence of binary predictions of whether each pitch is conditioned on lower-frequency pitches.

## 5 MODELS AND WEIGHT-SHARING

**Homophonic models** Factoring the score as discussed in Section 4 allows us to think of the polyphonic composition problem as a collection of coupled homophonic composition problems. We therefore consider a homophonic composition task on the KernScores dataset's parts: this is a single-part prediction task that generalizes classic monophonic prediction tasks to allow for chords. We explore a variety of fully connected, convolutional, and recurrent models for this task and find that the recurrent architecture works quite well. Table 5 summarizes these experiments; the third block of Experiments (15-21) compare the most interesting architectures over a reasonable amount of history.

The remaining question is: how do we encode the history of a polyphonic score, and how do we model correlations between parts in this history? We encode the history of a score as an order-3 binary tensor $\mathbf{x} \in \{0,1\}^{T \times P \times (N+D)}$, indexed by a time axis of length $T$, a part axis of length $P$, and an $(N+D)$-dimensional feature axis consisting of $N$-dimensional multi-hot vector of notes, and a $D$-dimensional one-hot vector of durations. The note bits $\mathbf{x}_{t,p,n}$ for $n \in \{0, \ldots, N-1\}$ indicate whether note $n$ is on at time $t$ in part $p$. The duration bits $\mathbf{x}_{t,p,d}$ for $d \in \{N, N+D-1\}$ indicate the duration of a homophonic event in part $p$ beginning at time $t$ or, if the previous state of part $p$ continues at time $t$, a special continuation duration '$*$' is indicated. In this way, we interlace the events in each part between each other: see Figure 2 for a visual example of this encoding.

Our hope is that a recurrent network designed for the single-part task would be relatively unhampered when retrained on part data interspersed with these continuations; Experiment 22 in Table 5 suggests that this is the case; performance degrades slightly in comparison to Experiment 21, but this is to be expected because a length-10 history interspersed with continuations is effectively a slightly shorter history. This encourages us to consider the coupled recurrent models described below.

**Polyphonic models** We now consider the full polyphonic task; results for this task are summarized in Table 3. We must now model correlations between parts in the history tensor, which we achieve coupling the representations of the individual parts. One natural extension of a recurrent neural network part model to multiple concurrent parts is a hierarchical architecture:

$$h_{p,t}(x_p) = \mathbf{a} \left( W_p^\top h_{p,t-1}(x_p) + W_x^\top x_{p,t} \right), \tag{2}$$

$$h_t(x) = \mathbf{a} \left( W_h^\top h_{t-1}(x) + W_{hp}^\top \sum_q h_{q,t}(x_q) \right). \tag{3}$$

Equation 2 is a recurrent network that builds a state estimate $h_{p,t}$ of an individual part $x_p$ at time $t$ based on transition weights $W_p$, emission weights $W_x$, and non-linear activation $\mathbf{a}$. Equation 3 is

---

[4]For polyphonic instruments like the piano, we must adopt a more refined definition of a part than "notes assigned to a particular instrument;" see Appendix A for details.

| 1.50 : 70 | 1.00 : 58 | 1.00 : 62 | 1.00 : 58 |
|-----------|-----------|-----------|-----------|
| * | 1.00 : | 1.00 : | 1.00 : 55 |
| 0.25 : 72 | * | * | * |
| 0.25 : 74 | * | * | * |
| 0.25 : 75 | 1.00 | 0.50 : | 1.00 : 51 |
| 0.25 : 77 | * | * | * |
| 0.25 : 79 | * | 0.25 : 75 | * |
| 0.25 : 81 | * | 0.25 : 77 | * |
| 1.00 : 82 | 3.00 : | 1.00 : 79 | 1.50 : 46 |
| ... | ... | ... | ... |

Figure 2: Left: a flattened description of the history tensor $\mathbf{x}_{t,p,n}$ of features for a Haydn string quartet: opus 55 number 3, first movement, from measure 16. Parts are indicated by columns. A frame of time is indicated by a row. Each event in a part is denoted by a pair of duration and note(s), separated by a colon. Durations are denominated in beats. An asterisk indicated continuation of the previous note(s). Right: the score corresponding to the history tensor (in a score, the time and part axes are transposed).

a global network that integrates the states of the part (weights $W_{hp}$) with the previous global state (weights $W_h$) to build a coupled global state $h_t$ at time $t$. Because the order of the parts is arbitrary, we sum over their states before feeding them into the global network. At each time step, we can use the learned state of each part together with the global state to predict what follows (see Figure 1). Another natural extension of a recurrent part model is to directly integrate the state of the other parts' states into each individual part's state, resulting in a distributed state architecture:

$$ h_{p,t}(x_p) = \mathrm{a}\left( W_p^\top h_{p,t-1}(x_p) + W_x^\top x_{p,t} + W_{hp}^T \sum_q h_{q,t}(x_q) \right). \tag{4} $$

We find that the distributed architecture underperforms the hierarchical architecture (see Table 3; Experiments 2 and 3) although this comparison is not conclusive: for example, the hierarchical model's implementation has more parameters than the distributed model. For the hierarchical model, we can also consider whether the global state representation is as sensitive to history-length as the parts. Could we make successful predictions using only the final state of each part, rather than coupling the states at each step? Experiments (4,5,6) in Table 3 suggest that this is not the case.

In the remainder of this section, we explore a variety of weight-sharing ideas that are somewhat orthogonal to our methods for factoring and modeling scores. These ideas may be of general interest for both monophonic and polyphonic composition, beyond the specific models under consideration.

## 5.1 AUTOREGRESSIVE MODELING

To build a generative model over sequential data $\mathbf{x} = (x_1, \ldots, x_t)$, rather than directly model the distribution $p(\mathbf{x})$, it often makes sense to factor the joint distribution into conditionals $p(x_t|x_{1:t})$ and make an autoregressive assumption $p(x_t|x_{1:t}) = p(x_s|x_{1:s})$ for all $s, t \in \mathbb{N}$. We can then learn a single conditional distribution $p(x_t|x_{1:t})$ and share model parameters across all time translations. If the data is conditionally stationary (or nearly so) this approach is extremely effective (analogous to convolution for vision problems).

Scores are not quite conditionally stationary; their distribution varies substantially depending on the position within the beat. For example, the distribution has quite a lot of variance on the beat and new notes are frequently introduced. In contrast, notes almost never begin an $\varepsilon$-fraction of time after the beat and the distribution is quite peaked on the notes initiated in the previous time-step. To address this non-stationarity, we follow the lead of Johnson (2017) and Hadjeres et al. (2017) and augment our history tensor with a one-hot location feature vector ($\ell$ in Table 5) that indicates the subdivision of the beat for which we are presently making predictions.[5] Compare the loss of duration models ($\mathrm{Loss}_t$) with and without these features in Experiment pairs (3,4), (6,7), (10,11), (12,13), and (15,16)

---

[5]Note that this location can always be computed from the full history tensor. But in practice we will truncate the history, effectively imposing a Markov assumption on our models and losing this information.

## 5.2    PART DECOMPOSITION

We have previously discussed decomposing a score into multiple parts. This presents us with an opportunity to share weights between part models by imposing the assumption $p(x_{t,p}|\mathbf{x}_{t,1:p}, \mathbf{x}_{1:t}) = p(x_{t,q}|\mathbf{x}_{t,1:q}, \mathbf{x}_{1:t})$ for all parts $p, q$. This corresponds to learning a single set of weights $W_p$ in equations (2) and (4), rather than learning unique part-indexed weights $W_{p_i}$ for each part $p_i$. Indeed, because the index of a part is arbitrary, the weights $W_{p_i}$ should converge to the same values for all $i$; sharing a single set of weights $W_p$ accelerates learning by enforcing this property.

## 5.3    RELATIVE PITCH

With a little effort, we can perform a similar weight-sharing scheme over the notes as we did over time and parts. This idea was first proposed in Johnson (2017). Recall that we factor the vector of pitches in part $p$ at time $t$ into a sequence of binary predictions, from lowest to highest pitch. Instead of building an individual predictor for each pitch conditioned on the notes in the history tensor, we can build a single predictor that conditions on a shifted version of the history tensor centered around the note we want to predict. By convolving this predictor over the pitch axis of the history tensor, we can make a prediction at each note location based on a relativized view of the history: see Figure 3 for a visualization of this transformation.

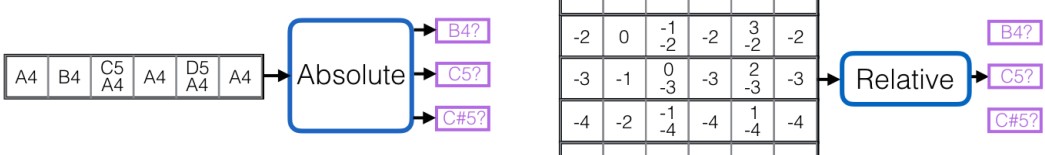

Figure 3: Left: an absolute pitch predictor that learns individual classifiers for each pitch-class. Right: a relative pitch predictor, that learns a single classifier and translates the data along the frequency axis to center it around the pitch to be predicted. Whereas the absolute predictor decides whether C5 is on given the previous note was A4, the relative predictor decides whether the note under consideration is on given the previous note was 3 steps below it.

Like with the time axis, we observe that the distribution over notes is not quite conditionally stationary along the note-class axis. For example, a truly relative predictor would generate notes uniformly across the note-class axis, whereas the actual distribution of notes concentrates around middle C. Therefore we augment our history tensor with a one-hot pitch-class feature vector $\mathbf{1}_n$ that indicates the note $n$ for which we are presently making a prediction. This allows us to take full advantage of all available information when making a prediction, while borrowing strength from shared harmonic patterns in different keys or octaves. We compare absolute pitch-indexed classifiers ($\mathbf{lin}_n$) to a single, relative pitch classifier ($\mathbf{lin}$) in Table 5: compare the loss of pitch models ($\text{Loss}_n$) in Experiments (2,3,4), (5,6,7), (8,9,10), (11,12,13), and (15,16).

## 5.4    PITCH EMBEDDINGS

Borrowing the concept of a word embedding from natural language processing, we consider learned embeddings of pitch vectors, denoted by $\mathbf{c}$ in Table 5. For recurrent models, we do not observe substantial performance benefits to learning these embeddings: compare Experiments (20,21) in Table 5. However, we do find that we can learn quite compact embeddings (16 dimensions for the experiments presented in this paper) without sacrificing performance, and working on these compact embeddings speeds up processing time for learning and generation. We also find that a simple 12 dimensional fixed embedding of pitches $\mathbf{f}$, in which we quotient each pitch class by octave, reduces overfitting for the rhythmic model while preserving performance.

## 6    EXPERIMENTAL RESULTS

**Quantitative Results.** The single-part (homophonic) and multi-part (polyphonic) prediction tasks are presented in tables 5 and 3 respectively. We advise caution in thinking about these results. Small

differences in the log-loss can have large effects on the quality of output, especially if the differences are attributable to missing information. For example, failing to include a pitch-class feature vector in the relative models (described in Section 5.3) has a catastrophic impact on generated sequences, even though the log loss gap between is not always large. Detailed discussions of the individual results in these tables are presented in context in Section 5.

| # | History (part/global) | Architecture | Loss (total) | $\text{Loss}_t$ (time) | $\text{Loss}_n$ (notes) |
|---|---|---|---|---|---|
| 1 | 3 / 3 | hierarchical | 14.05 | 5.65 | 8.40 |
| 2 | 5 / 5 | hierarchical | 13.40 | 5.35 | 8.04 |
| 3 | 5 | distributed | 13.82 | 5.41 | 8.41 |
| 4 | 10 / 1 | hierarchical | 13.20 | 5.22 | 7.98 |
| 5 | 10 / 5 | hierarchical | 12.94 | 5.13 | 7.81 |
| 6 | 10 / 10 | hierarchical | 12.87 | 5.12 | 7.75 |
| 7 | 20 / 20 | hierarchical | 12.78 | 5.01 | 7.76 |
| 8 | 10 | independent | 18.63 | 6.56 | 12.08 |

Table 3: Multi-part results. The "hierarchical" architecture is defined by equations (2) and (3), and the "distributed" architecture is defined by equation (4): see the polyphonic models discussion in Section 5. Part and global history refer to the number of time steps used to construct the part states $h_{p,t}$ and global states $h_t$ respectively. Experiment 8 is a baseline in which the part models are completely decoupled (this is equivalent to single-part Experiment 22 in Table 5; on average, a full score in the KernScores dataset has $4.12$ parts). Results are reported on non-piano test set data (see Appendix A for discussion of piano data).

**Qualitative results** We asked twenty study participants to listen to a variety of audio clips, synthesized from either a real composition or from the output of one of our models: Experiment 4 in Table 3. For each clip, participants were asked to rate whether they thought the clip was written by a computer or by a human composer. Participants were presented with clips of varying length, from 10 frames of data (2-3 seconds; the length of the model's Markov window) to 50 frames of data (10 or more seconds). We expected that participant success would improve with the length of the clips, but we did not find this to be the case; indeed, even among the longest clips (around 20 seconds) participants occasionally identified an artificial clip as a human composition. Results are presented in Table 4.

| Clip Length | 10 | 20 | 30 | 40 | 50 |
|---|---|---|---|---|---|
| Average | 5.78 | 6.89 | 6.61 | 6.72 | 6.78 |

Table 4: Qualitative evaluation of the 10-frame hierarchical model: Experiment 4 in Table 3. Twenty participant were asked to judge 50 audio clips each of varying length. The scores indicate participants' average correct discriminations out of 10 (5.0 would indicate random guessing; 10.0 would indicate perfect discrimination).

As with the quantitative results, we again urge caution in interpreting these qualitative results. Our study results superficially suggest that we have done well in modeling the short-term structure of our dataset (we make no claims to have captured long-term structure; indeed, the Markov windows of our models preclude this). But it is not clear that humans are good (or should be good) at the task of identifying plausible or implausible local structure in music. It is also unclear how to use such studies to compare between models, where differences would be less pronounced. Indeed, it is not even clear how to prompt a user to discriminate in such a setting.

## 7 CONCLUSION

Given the difficulties evaluation generative models, it may be important to pay further attention to downstream tasks. One important downstream task is music transcription, which is considered together with polyphonic composition in Boulanger-Lewandowski et al. (2012) and more recently in

Sigtia et al. (2016). Both these systems operate on performance-aligned label sequences: a warping of a score to an expressive performance. More work would be necessary to generate an actual score that correctly identifies the value of notes (e.g. quarter note, or half-note) and not just the durations of notes in the audio.

The authors would also like to point out the oddity of training generative models on such a diverse set of music: from Josquin to Joplin. Music is inherently a low-resource learning problem; for comparison, modern language models are regularly trained on datasets larger than the entire classical music canon (Chelba et al., 2014). Fortunately, music has a much lower entropy rate than language. But we may need new tools to learn properly to compose in the style of Mozart.

| # | Params | Model | Loss | $\text{Loss}_t$ | $\text{Loss}_n$ |
|---|---|---|---|---|---|
| 1 | 112 | $\hat{y}_t = \mathbf{bias}_t, \hat{y}_n = \mathbf{bias}_n$ | 10.07 | 2.92 | 7.15 |
| 2 | 21k | $\hat{y}_t = \mathbf{lin}(x_1), \hat{y}_n = \mathbf{lin}_n(x_1, y_t, y_{1:n})$ | 8.05 | 2.00 | 6.05 |
| 3 | 9k | $\hat{y}_t = \mathbf{lin}(x_1), \hat{y}_n = \mathbf{lin}(x_1, y_t, y_{1:n})$ | 6.29 | 2.00 | 4.29 |
| 4 | 11k | $\hat{y}_t = \mathbf{lin}(x_1, \ell), \hat{y}_n = \mathbf{lin}(x_1, y_t, y_{1:n}, \mathbf{1}_n)$ | 6.12 | 1.83 | 4.29 |
| 5 | 149k | $\hat{y}_t = \mathbf{lin} \circ \mathbf{fc}(x_1), \hat{y}_n = \mathbf{lin}_n \circ \mathbf{fc}(x_1, y_t, y_{1:n})$ | 5.92 | 1.99 | 3.93 |
| 6 | 135k | $\hat{y}_t = \mathbf{lin} \circ \mathbf{fc}(x_1), \hat{y}_n = \mathbf{lin} \circ \mathbf{fc}(x_1, y_t, y_{1:n})$ | 6.05 | 1.99 | 4.07 |
| 7 | 172k | $\hat{y}_t = \mathbf{lin} \circ \mathbf{fc}(x_1, \ell), \hat{y}_n = \mathbf{lin} \circ \mathbf{fc}(x_1, y_t, y_{1:n}, \mathbf{1}_n)$ | 5.70 | 1.80 | 3.90 |
| 8 | 72k | $\hat{y}_t = \mathbf{lin}(x_5), \hat{y}_n = \mathbf{lin}_n(x_5, y_t, y_{1:n})$ | 7.91 | 1.86 | 6.05 |
| 9 | 36k | $\hat{y}_t = \mathbf{lin}(x_5), \hat{y}_n = \mathbf{lin}(x_5, y_t, y_{1:n})$ | 5.76 | 1.86 | 3.91 |
| 10 | 38k | $\hat{y}_t = \mathbf{lin}(x_5, \ell), \hat{y}_n = \mathbf{lin}(x_5, y_t, y_{1:n}, \mathbf{1}_n)$ | 5.63 | 1.73 | 3.91 |
| 11 | 418k | $\hat{y}_t = \mathbf{lin} \circ \mathbf{fc}(x_5), \hat{y}_n = \mathbf{lin}_n \circ \mathbf{fc}(x_5, y_t, y_{1:n})$ | 4.90 | 1.64 | 3.26 |
| 12 | 497k | $\hat{y}_t = \mathbf{lin} \circ \mathbf{fc}(x_5), \hat{y}_n = \mathbf{lin} \circ \mathbf{fc}(x_5, y_t, y_{1:n})$ | 4.80 | 1.64 | 3.16 |
| 13 | 535k | $\hat{y}_t = \mathbf{lin} \circ \mathbf{fc}(x_5, \ell), \hat{y}_n = \mathbf{lin} \circ \mathbf{fc}(x_5, y_t, y_{1:n}, \mathbf{1}_n)$ | 4.68 | 1.59 | 3.10 |
| 14 | 228k | $\hat{y}_t = \mathbf{lin} \circ \mathbf{fc}(\mathbf{f}(x_5), \ell)$ 
 $\hat{y}_n = \mathbf{lin} \circ \mathbf{fc}(\mathbf{c}(x_5), y_t, y_{1:n}, \mathbf{1}_n)$ | 4.63 | 1.58 | 3.05 |
| 15 | 134k | $\hat{y}_t = \mathbf{lin}(x_{10}), \hat{y}_n = \mathbf{lin}_n(x_{10}, y_t, y_{1:n})$ | 7.88 | 1.83 | 6.05 |
| 16 | 71k | $\hat{y}_t = \mathbf{lin}(x_{10}, \ell), \hat{y}_n = \mathbf{lin}(x_{10}, y_t, y_{1:n}, \mathbf{1}_n)$ | 5.53 | 1.71 | 3.83 |
| 17 | 372k | $\hat{y}_t = \mathbf{lin} \circ \mathbf{fc}(\mathbf{f}(x_{10}), \ell)$ 
 $\hat{y}_n = \mathbf{lin} \circ \mathbf{fc}(\mathbf{c}(x_{10}), y_t, y_{1:n}, \mathbf{1}_n)$ | 4.55 | 1.55 | 3.00 |
| 18 | 250k | $\hat{y}_t = \mathbf{lin} \circ \mathbf{conv}_5(\mathbf{f}(x_{10}), \ell)$ 
 $\hat{y}_n = \mathbf{lin} \circ \mathbf{conv}_5(\mathbf{c}(x_{10}), y_t, y, \mathbf{1}_n)$ | 4.56 | 1.55 | 3.01 |
| 19 | 769k | $\hat{y}_t = \mathbf{lin} \circ \mathbf{conv}_3 \circ \mathbf{conv}_5(\mathbf{f}(x_{10}), \ell)$ 
 $\hat{y}_n = \mathbf{lin} \circ \mathbf{conv}_3 \circ \mathbf{conv}_5(\mathbf{c}(x_{10}), y_t, y_{1:n}, \mathbf{1}_n)$ | 4.42 | 1.50 | 2.92 |
| 20 | 342k | $\hat{y}_t = \mathbf{lin} \circ \mathbf{rnn}(x_{10}, \ell)$ 
 $\hat{y}_n = \mathbf{lin} \circ \mathbf{rnn}(x_{10}, y_t, y_{1:n}, \mathbf{1}_n))$ | 4.37 | 1.48 | 2.89 |
| 21 | 283k | $\hat{y}_t = \mathbf{lin} \circ \mathbf{rnn}(\mathbf{f}(x_{10}), \ell)$ 
 $\hat{y}_n = \mathbf{lin} \circ \mathbf{rnn}(\mathbf{c}(x_{10}), y_t, y_{1:n}, \mathbf{1}_n))$ | 4.36 | 1.48 | 2.88 |
| 22 | 301k | $\hat{y}_t = \mathbf{lin} \circ \mathbf{rnn}(\mathbf{f}(\tilde{x}_{10}), \ell)$ 
 $\hat{y}_n = \mathbf{lin} \circ \mathbf{rnn}(\mathbf{c}(\tilde{x}_{10}), y_t, y_{1:n}, \mathbf{1}_n))$ | 4.52 | 1.59 | 2.93 |

Table 5: Single-part results. Loss is the cross-entropy described in Section 3.1. $\text{Loss}_t$ and $\text{Loss}_n$ are decompositions of the loss into component losses for duration $\hat{y}_t$ and pitch $\hat{y}_n$ predictions respectively. $\mathbf{lin}_n$ indicates a log-linear classifier (sigmoid for $\hat{y}_n$ and softmax for $\hat{y}_t$). The inclusion of location features discussed in Section 5.1 is indicated by $\ell$. $\mathbf{lin}$ indicates the relative pitch log-linear classifier described in Section 5.3 and $\mathbf{1}_n$ indicates the inclusion of pitch-class features. $\mathbf{fc}$ indicates a fully connected layer. $\mathbf{f}$ and $\mathbf{c}$ indicates the pitch embeddings described in Section 5.4. $\mathbf{conv}_k$ indicates 1d convolution of width $k$. $\mathbf{rnn}$ indicates a recurrent layer. All hidden layers are parameterized with 300 nodes. Models were regularized with early stopping when necessary. The subscript $k$ on the history tensor $x_k$ indicates the number of frames of history used in each experiment (either 1, 5, or 10 frames). The history tensor $\tilde{x}_k$ is modified to include continuation symbols '*' as it would in the polyphonic prediction task; see the discussion at the top of section 5.

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

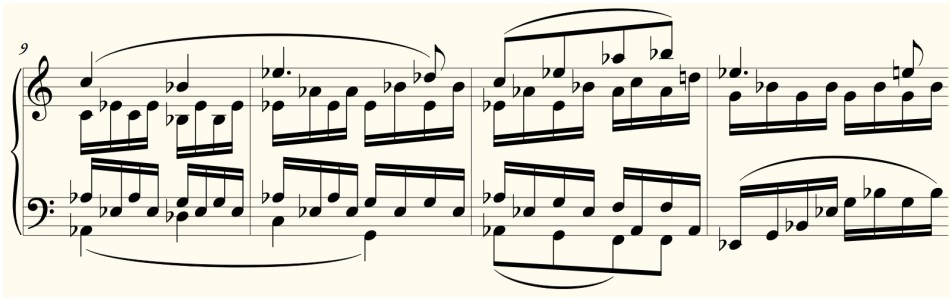

Figure 4: Beethoven's piano sonata number 8 (Pathetique) movement 2, from measure 9, rendered by the Verovio Humdrum Viewer. Although visually rendered on two staves, this sonata consists of four parts: a high sequence of quarter and eighth notes, two middle sequences of sixteenth notes, and a low sequence of quarter notes.

## A    Piano Music

For piano music, we need to draw a distinction between an instrument and a part. Consider the piano score given in Figure 4. This single piano part is more comparable to a complete score than the individual parts of, for example, a string quartet (compare the piano score in Figure 4 to the quartet score in Figure 2). Indeed, an educated musician would read this score in four distinct parts: a high sequence of quarter and eighth notes, two middle sequences of sixteenth notes, and a low sequence of quarter notes. In measure 12, the lowest two parts combine into a single bass line of sixteenth notes.

These part divisions are indicated in score through a combination of beams, slurs, and other visual queues. We do not model these visual indicators; instead we rely on part annotations provided by the KernScores dataset. The provision of these annotations is a strong point in favor of the KernScores dataset's Humdrum format; although in principle formats like MIDI can encode this information, in practice they typically collect all notes for a single instrument into a single track, or possibly two tracks (for the treble and bass staves, as seen in the figure) in the case of piano music.

In extremely rare cases, this distinction between instrument and part must also be made for stringed instruments; a notable example is Beethoven's string quartet number 14, in the fourth movement in measures 165 and 173, where the four instruments each separate into two distinct parts creating brief moments of 8-part harmony. The physical constraints of stringed instruments discourage more widespread use of these polyphonies. For vocal music, of course, physical constraints prevent intra-instrument polyphony entirely.

As Figure 4 illustrates, these more abstract parts can weave in and out of existence. Two parts can merge with each other; a single part can split in two; new parts can emerge spontaneously. The KernScores data provides annotations that describe this behavior. We can represent these dynamics of parts as a $P \times P$ flow matrix at each time step (where $P$ is an upper bound on the number of parts; in our corpus $P = 6$) that describes where each part moves in the next step. At most time steps, this flow matrix is the identity matrix.

The state-based models discussed in this paper can easily be adjusted to accommodate these flows. If two parts merge, sum their states; if a part splits in two, duplicate its state. These operations amount to hitting the vector of state estimates for the parts with the flow matrix at each time step. However, we do not currently model the flow matrix. Because the flow matrix for piano music contains some (small) amount of entropy, we therefore exclude piano music from the results reported in Table 3. We do however include the piano music in training.

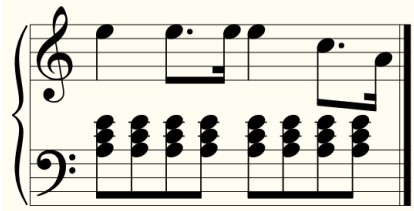

Figure 5: Mozart's piano sonata number 8 in A minor, movement 1, from measure 1, rendered by the Verovio Humdrum Viewer.

## B    SEVERAL WAYS TO FACTOR THE DISTRIBUTION OVER SCORES

As discussed in section 4, we want to order a factor a score over notes. A score consists of notes, each of which has a start time, part assignment, duration, and pitch. Formally, we can think of a score as a list of notes $S$, where each note is a tuple $(s, p, n, d)$ where $s \in \mathbb{R}_+$ is the start time of the note (units of beats), $p \in \{1, \ldots, P\}$ is the part assignment of the note, $n \in \{1, \ldots, N\}$ is the pitch of the note (e.g. MIDI code), and $d \in \mathbb{R}_+$ is the duration of the note (units of beats).

We would like to factor the distribution over scores, to take advantage of its autoregressive nature and other structure. We are immediately faced with the question of how to handle continuous time.

### B.1    DISCRETE "RASTER" FACTORIZATION

One approach to making predictions in a continuous domain is to discretize the domain. As discussed in Section 5, we can choose a discrete time step $\Delta > 0$ such that $s_i = k_i \Delta$ and $d_i = j_i \Delta$ with $k \in \mathbb{N}$ for all times $s_i$ and durations $d_i$ in all scores. In other words, we can discretize a score without losing information: we just need to specify its contents at each point on a discrete grid of resolution $\Delta$. To this end we define a discretized score $\mathbf{x} \in \{0, 1\}^{T \times P \times 2N}$ where, at each time step $t \in \{1, \ldots, T\}$, for each part $p \in \{1, \ldots, P\}$ and each note $n \in \{1, \ldots, N\}$, we set

$$\mathbf{x}_{t,p,n} = 1 \qquad\qquad \text{iff note } n \text{ is on at time } t \text{ in part } p,$$
$$\mathbf{x}_{t,p,2n} = 1 \qquad\qquad \text{iff note } n \text{ begins at time } t \text{ in part } p.$$

Observe that both a "note" bit and an "onset" bit are required to faithfully represent the score $S$; with only a single bit to indicate the occurrence of a note, both the scores in Figure 7 would be confused with the score in Figure 5.

We can then factor the distribution over scores into a binary-valued conditional distributions:

$$p(\mathbf{x}) = \prod_{t=1}^{T} p(\mathbf{x}_t | \mathbf{x}_{1:t}) = \prod_{t=1}^{T} \prod_{p=1}^{P} p(\mathbf{x}_{t,p} | \mathbf{x}_{1:t}, \mathbf{x}_{t,1:p}) = \prod_{t=1}^{T} \prod_{p=1}^{P} \prod_{n=1}^{2N} p(\mathbf{x}_{t,p,n} | \mathbf{x}_{1:t}, \mathbf{x}_{t,1:p}, \mathbf{x}_{t,p,1:n}).$$

Discretization is the approach taken by Boulanger-Lewandowski et al. (2012), with some caveats. First, their work doesn't model parts (this can be recovered in our exposition by setting $P = 1$). Second, their work doesn't take the final step of factoring $p(\mathbf{x}_{t,p} | \mathbf{x}_{1:t}, \mathbf{x}_{t,1:p})$ into binary-valued distributions; they choose rather to directly target the vector $\mathbf{x}_t$ with NADE. Third, their work using a single bit to indicate notes, thus losing expressivity in the manner discussed in Figure 7. Fourth, rather than work with the process resolution $\Delta$, their work discretizes at a coarse resolution of eighth notes (quarter notes in the case of the JSB chorales segment of their corpus) thus losing substantial information. Figure 6 illustrates the effect of discretizing the score in Figure 5 at eighth-note resolution.

### B.2    RUN-LENGTH FACTORIZATION

Training and sampling from a model over a discrete factorization of scores at the process resolution $\Delta$ can be expensive, prompting some earlier works to discretize at a coarser resolution (as discussed

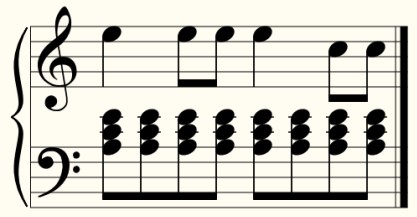

Figure 6: A visualization of the score from Figure 5, discretized at eighth-note resolution.

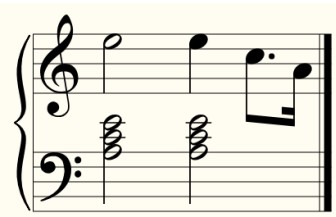

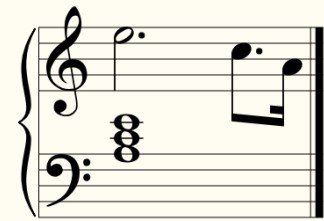

Figure 7: Two scores with the same raster representation as the score in Figure 5, if only a single bit is used to indicate the occurrence of note $n$ at time $t$. Other variants of these scores with the same raster encoding are also possible. The popular dataset introduced by Boulanger-Lewandowski et al. (2012) uses a single-bit raster. A second bit is used in some more recent work, for example Liang et al. (2017) in which they are referred to as "Tie" bits).

in the previous section). One approach to preserve fine rhythmic structure (e.g. triplets and thirty-second notes) without committing to a fine discretization is to factor a score into run-lengths. To this end, we define a run-length encoded score $\mathbf{x} \in \left(\mathbb{N} \oplus \{0,1\}^{P \times 2N}\right)^T$ where, at each time index $t \in \{1, \ldots, T\}$, we set

$$\mathbf{x}_{t,0} = \mathbf{1}_{d_t}, \qquad \text{where } d_t \text{ is the duration of the event at time index } t,$$
$$\mathbf{x}_{t,1,p,n} = 1 \qquad \text{iff note } n \text{ is on at time } t \text{ in part } p,$$
$$\mathbf{x}_{t,1,p,2n} = 1 \qquad \text{iff note } n \text{ begins at time } t \text{ in part } p.$$

Crucially, the sequence $\mathbf{x}_t$ is non-linear in the index $t$: entry $\mathbf{x}_{t+1}$ occurs $d_t$ beats after entry $\mathbf{x}_t$, in contrast to the raster where $\mathbf{x}_{t+1}$ always occurs a constant interval $\Delta$ after $\mathbf{x}_t$.

We can then factor the distribution over scores into conditional distributions over binary note values and natural-number duration values:

$$p(S) = \prod_{t=1}^{T} p(\mathbf{x}_t | \mathbf{x}_{1:t}) = \prod_{t=1}^{T} p(\mathbf{x}_{t,0} | \mathbf{x}_{1:t}) \prod_{p=1}^{P} \prod_{n=1}^{2N} p(\mathbf{x}_{t,p,n} | \mathbf{x}_{1:t}, \mathbf{x}_{t,0}, \mathbf{x}_{t,1,1:p}, \mathbf{x}_{t,1,p,1:n}).$$

Because music typically doesn't evolve at the finest possible resolution $\Delta$, we save a substantial amount of computation by predicting run-lengths $\mathbf{x}_{t,0} \in \mathbb{N}$ rather than re-iterating the predictions $\mathbf{x}_{t,p,n}$ at successive time steps.

One criticism of the run-length factorization is that, when notes of different durations overlap in a score, the longer notes are chopped up along the boundaries of the short notes as illustrated in Figure 8. Instead of predicting musically meaningful quantities like note values (quarter, eighth, dotted-eighth, etc.) instead we predict run-length chunks.

### B.3 OPERATIONALIZED FACTORIZATION

Both the raster and run-length factorizations discussed above don't directly recognize the concept of note-values: quarter-notes, half-notes, eighth-notes, etc.. One approach to factorization that explicitly models note-values is to describe a score as a sequence of operations that produce the score, and then factor this sequence. Formally, we can describe an operationalized score as a sequence

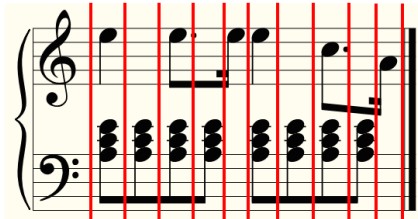

Figure 8: The Mozart from Figure 5, with red lines that indicate the boundaries of events under a run-length factorization of the score. Notes in the treble staff are chopped up into eight-note runs, so instead of predicting note durations (quarter, dotted-eighth, sixteenth, etc.) we instead predict fragments of notes (eighth, continue eighth, continue eighth, etc.).

$\mathbf{x} \in (\mathbb{N} \oplus (\mathbb{N} \oplus \{0,1\}^{P \times N}))^L$ of length $L$, where each index $\mathbf{x}_i$ is three-hot with the following semantics:

| | |
|---|---|
| $\mathbf{x}_{i,0,d} = 1$ | $\mathbf{x}_i$ is a time-shift operation: time advances by $d$ steps, |
| $\mathbf{x}_{i,1,v} = 1$ and $\mathbf{x}_{i,2,n} = 1$ | $\mathbf{x}_i$ is a note operation: note $n$ with note-value $v$ begins playing. |

This approach explicitly learns to predict note values $v$.

We remark that there are other operational factorizations that capture note values implicitly, like the raster or run-length encodings. Another factorization, similar to Oore et al. (2018) and inspired by the popular MIDI format, operationalizes a score as a sequence $\mathbf{x} \in (\mathbb{N} \oplus \{0,1\}^{P \times N} \oplus \{0,1\}^{P \times N})^L$ of length $L$, where each index $\mathbf{x}_i$ is one-hot with the following semantics:

| | |
|---|---|
| $\mathbf{x}_{i,0,d} = 1$ | $\mathbf{x}_i$ is a time-shift operation: time advances by $d$ steps, |
| $\mathbf{x}_{i,1,p,n} = 1$ | $\mathbf{x}_i$ is a note-on operation: note $n$ begins playing in part $p$, |
| $\mathbf{x}_{i,2,p,n} = 1$ | $\mathbf{x}_i$ is a note-off operation: note $n$ stops playing in part $p$. |

In either case, we can then factor the distribution over these sequences:

$$p(S) = \prod_{i=1}^{L} p(\mathbf{x}_i | \mathbf{x}_{1:i}).$$

While the first operational approach describes above explicitly predicts note values, both operational factorizations require us to predict time-shifts, which are analogous to the durations predicting in the run-length factorization described in Appendix B.2. We will now turn our attention to a single-part setting where run-lengths are equivalent to note values, thus eliminating these strange intermediate quantities.

### B.4 Run-Length Factorization for Parts

Observe in Figure 8 that if we were to run-length encode or operationalize the top and bottom staff individually, the run-lengths or time-shifts respectively would correspond to note-values. Because these parts are individually homophonic–notes within a part move in rhythmic lock-step–we do not observe the phenomenon where short notes overlap long notes in time, resulting in chopped up run-lengths. Applying the run-length factorization described in Appendix B.2 we see that, in the context of homophonic parts, run-lengths are equivalent to note-values and therefore the run-length factorization explicitly models these note values. The KernScores dataset takes this homophony property as the definition of a part (they call parts "voices") and therefore encode the score in Figure 5 as two parts; in cases of polyphony as in Figure 4, the KernScores dataset decomposes the score into additional parts to maintain the homophony property.

Therefore, we can construct a run-length factorization of the KernScores parts that explicitly models note-values. We describe a part by $\mathbf{x} \in (\mathbb{N} \oplus \{0,1\}^N)^T$ where, at each time index $t \in \{1, \ldots, T\}$,

we set

$$\mathbf{x}_{t,0} = \mathbf{1}_{d_t}, \qquad \text{where } d_t \text{ is the duration of the note (or chord) at time index } t,$$
$$\mathbf{x}_{t,1,n} = 1 \qquad \text{iff note } n \text{ is on at time } t.$$

In contrast to the run-length encoding described in Appendix B.2, we no longer need a second bit to distinguish onsets from continuations; because notes don't get chopped up, we will never need a bit to indicating that we are re-iterating a note that was initiated in a previous run. We can then proceed to factor the part as:

$$p(\mathbf{x}) = \prod_{t=1}^{T} p(\mathbf{x}_{t,0}|\mathbf{x}_{1:t}) \prod_{n=1}^{N} p(\mathbf{x}_{t,n}|\mathbf{x}_{1:t}, \mathbf{x}_{t,0}, \mathbf{x}_{t,1,1:n}).$$

Learning the conditional distributions of pitches $p(\mathbf{x}_{t,n}|\mathbf{x}_{1:t}, \mathbf{x}_{t,0}, \mathbf{x}_{t,1,1:n})$ and note-values $p(\mathbf{x}_{t,0}|\mathbf{x}_{1:t})$ is the **homophonic modeling** task considered in Section 5.

## B.5 INTERLACING PARTS TO FACTOR A SCORE

Let $S$ be a score consisting of $P$ parts. The run-lengths that comprise each part can be a totally ordered by the timestamp each run-length begins. We extend this to a total order across parts using these same timestamps, breaking ties (where run-lengths begin at the same time in multiple parts) using the arbitrary given by the indices of the parts. For an index $i$ in this total order, let $\mathcal{P}_i$ be the part associated with this index and $\mathcal{T}_i$ be the time (by definition, $\mathcal{T}_i$ is a non-decreasing sequence). Finally, let $L_p$ be the number of run-lengths in part $p$ and define $L \equiv \sum_{p=1}^{P} L_p$.

For example, consider the score in Figure 5. In this case $L_0 = 6$ (the treble part) and $L_1 = 8$ (the bass part). The time $\mathcal{T}$ and part $\mathcal{P}$ index sequences are given by

$$\mathcal{T} = [0, \quad 0, \quad .5, \quad 1, \quad 1, \quad 1.5, \quad 1.75, \quad 2, \quad 2, \quad 2.5, \quad 3, \quad 3, \quad 3.5, \quad 3.75],$$
$$\mathcal{P} = [0, \quad 1, \quad 1, \quad 0, \quad 1, \quad 1, \quad 0, \quad 0, \quad 1, \quad 1, \quad 0, \quad 1, \quad 1, \quad 0].$$

We describe a score by $\mathbf{x} \in (\mathbb{N} \oplus \{0,1\}^N)^L$ where, at each index $\mathbf{x}_i$, we set

$$\mathbf{x}_{i,0} = \mathbf{1}_{d_i}, \qquad \text{where } d_i \text{ is the duration of the note (or chord) at time } \mathcal{T}_i \text{ in part } \mathcal{P}_i,$$
$$\mathbf{x}_{i,1,n} = 1 \qquad \text{iff note } n \text{ is on at time } \mathcal{T}_i \text{ in part } \mathcal{P}_i.$$

We then proceed to factor this distribution in the same manner as the homophonic parts model:

$$p(\mathbf{x}) = \prod_{i=1}^{L} p(\mathbf{x}_{i,0}|\mathbf{x}_{1:i}) \prod_{n=1}^{N} p(\mathbf{x}_{i,n}|\mathbf{x}_{1:i}, \mathbf{x}_{i,0}, \mathbf{x}_{i,1,1:n}).$$

This factorization models the next run in part $\mathcal{P}(i)$ as we did in the homophonic model, but additionally condition on runs in other parts that were initiated prior to the run we are currently predicting (and also on runs initiated at the same time in parts with indices less than $p$). Note that we do not need to predict $\mathcal{T}$ and $\mathcal{P}$; they are completely determined by the sequential ordering given on the $\mathbf{x}_i$'s. This is the **polyphonic modeling** task considered in Section 5.

