# OpenReview forum: "Coupled Recurrent Models for Polyphonic Music Composition"
_ICLR.cc/2019/Conference_

### Official Review · AnonReviewer2 · 2018-11-03
**sections almost made sense, output almost sounded good, ... ("4: Ok but not good enough")**

**Rating:** 4
**Confidence:** 4

**Review:**

PROs
-seemingly reasonable approach to polyphonic music generation: figuring out a way to splitting the parts, share parameters appropriately, measuring entropy per time, all make sense
-the resulting outputs tend to have very short-term harmonic coherence (e.g. often a ‘standard chord’ with some resolving suspensions, etc), with individual parts often making very small stepwise motion (i.e. reasonable local voice leading)
-extensive comparison of architectural variations
-positive results from listening experiments

CONs
-musical outputs are *not* clearly better than some of the polyphonic systems described; despite the often small melodic steps, the individual lines are quite random sounding; this is perhaps a direct result of the short history
-I do not hear the rhythmic complexity that is described in the introduction
-the work by Johnson (2015) (ref. provided below) should be looked at and listened to; it too uses coupled networks, albeit in a different way but with a related motivation, and has rhythmic and polyphonic complexity and sounds quite good (better, in my opinion)
-some unclear sections (fixable, especially with an appendix; more detail below)
-despite the extensive architectural comparisons, I was not always clear about rationale behind certain choices, eg. if using recurrent nets, why not try LSTM or GRU? (more questions below)
-would like to have heard the listening tests; or at least read more about how samples were selected (again, perhaps in an appendix and additional sample files)

 quality, clarity, originality and significance of this work, including a list of its pros and cons (max 200000 characters).

Quality -- In this work, various good/reasonable choices are made. The quality of the actual output is fine. It is comparable to-- and to my ears not better than-- existing polyphonic systems such as the ones below (links to sample audio are provided here):

-Bachbot - https://soundcloud.com/bachbot (Liang et al 2017)
- tied parallel nets - http://www.hexahedria.com/2015/08/03/composing-music-with-recurrent-neural-networks/ (Johnson 2015, ref below)
-performanceRNN - https://magenta.tensorflow.org/performance-rnn - (Simon & Oore 2017)
..others as well..


Clarity -- Some of the writing is "locally" clear, but one large, poorly-organized section makes the whole thing confusing (details below). It is very helpful that the authors subsequently added a comment with a link to some sample scores; without that, it had been utterly impossible to evaluate the quality. There are a few points that could be better clarified:
	-p5”a multi-hot vector of notes N”. It sounds like N will be used to denote note-numbers, but in fact it seems like N is the total number of notes, i.e. the length of the vector, right? What value of N is used?
-p5 “a one-hot vector of durations D”. It sounds like D will be used to denote durations, but actually I think D is the length of the 1-hot vector encoding durations right? What value of D is used, and what durations do the elements of this vector represent?
-similarly, does T represent the size of the history? This should really be clarified.
	-p5 Polyphonic models.
		-Eq (2), (3), (4): Presumably the h’s are the hidden activations layers?
		-the networks here correspond to the blue circles in Fig 1, right? If so, make the relationship clear and explicit
		-Note that most variables in most equations are left undefined
		-actually defining the W’s in Eq(2-4)  would allow the authors to refer to the W’s later (e.g. in Section 5.2) when describing weight-sharing ideas. Otherwise, it’s all rather confusing. For example, the authors could write, “Thus, we can set W_p1 = W_p2 = W_p3 = W_p4” (or whatever is appropriate).
	-Generally, I found that pages 5-7 describe many ideas, and some of them are individually fairly clearly described, but it is not always clear when one idea is beginning, and one idea is ending, and which ideas can be combined or not. On my first readings, I thought that I was basically following it, until I got to Table 5, which then convinced me that I was in fact *not* quite following it. For example, I had been certain that all the networks described are recurrent (perhaps due to Fig1?), but then it turned out that many are in fact *not* recurrent, which made a lot more sense given the continual reference to the history and the length of the model’s Markov window etc. But the reader should not have had to deduce this. For example, one could write,
	“We will consider 3 types of architectures: convolutional, recurrent, .... In each architecture, we will have [...] modules, and we will try a variety of combinations of these modules. The modules/components are as follows:”. It’s a bit prosaic, but it can really help the reader.
-Appendices, presented well, could be immensely helpful in clarifying the exact architectures; obviously not all 22 architectures from Table 5 need to be shown, but at least a few of them shown explicitly would help clarify. For example, in Fig1, the purple boxes seem to represent notes (according to the caption), but do they actually represent networks? If they really do represent notes, then how can “notes” receive inputs from both the part-networks and the global network? Also, I was not entirely clear on the relationship of the architecture of the individual nets (for the parts) to that of the global integrating network. E.g. for experiment #20, the part-net is an RNN (with how many layers?? with regular or LSTM cells?) followed by a log-linear predictor (with one hidden layer of 300 units right? or are there multiple layers sometimes?), but then what is the global network? Why does the longest part-history vector appear to have length 10 based on Table 5, but according to Table 3 the best-performing history length was 20? Though, I am not sure the meaning of the “bottom/top” column was explained anywhere, so maybe I am completely misunderstanding that aspect of the table? Etc.
-Many piano scores do not easily deconstruct into clean 4-part polyphony; the example in Appendix A is an exception. It was not clear to me how piano scores were handled during training.
-Terminology: it is not entirely clear to me why one section is entitled “homophonic models”, instead of just “monophonic models”. Homophonic music usually involves a melody line that is supported by other voices, i.e. a sort of asymmetry in the part-wise structure. Here, the outputs are quite the opposite of that: the voices are independent, they generally function well together harmonically, and there is usually no sense of one voice containing a melody. If there’s some reason to call it homophonic, that would be fine, but otherwise it doesn’t really serve to clarify anything. However, the authors do say that the homophonic composition tasks are a “minor generalization of classic monophonic composition tasks”, so this suggests to me that there is something here that I am not quite understanding.

The last sentence of Section 5.3 is very confusing-- I don’t understand what lin_n is, or 1_n is, or how to read the corresponding entries of the table. The first part of the paragraph is fairly clear.

Table 4: “The first row” actually seems like it is referring to the second row. I know what the authors mean, but it is unnecessarily confusing to refer to it in this way. One might as well refer to “the zeroth row” as listing the duration of the clip :)

The experimental evaluation: I would like to hear some of the paired samples that were played for subjects. Were classical score excerpts chosen starting at random locations in the score, or at the beginning of the score? It is known that listening to a 10-second excerpt without context can sometimes not make sense. I would be curious to see the false positives versus the false negatives. Nevertheless, I certainly appreciate the authors’ warning to interpret the listening results with caution.




Originality & Significance -- So far, based both on the techniques and the output, I am not entirely convinced of the originality or significance of this particular system. The authors refer to “rhythmically simple polyphonic scores” such as Bachbot, but I cannot see what is rhythmically fundamentally more sophisticated about the scores being generated by the present system. One nice characteristic of the present system is the true and audible independence of the voices.

One of the contributions appears to be the construction of models that explicitly leverage with shared weights some of the patterns that occur in different “places” (pitch-wise and temporally) in music. This is both very reasonable, and also not an entirely novel idea; see for example the excellent work by Daniel Johnson, “Generating Polyphonic Music Using Tied Parallel Networks” (paper published 2017, first shared online, as far as I know, in 2015: links to all materials available at http://www.hexahedria.com/2015/08/03/composing-music-with-recurrent-neural-networks/  )
Another now common (and non exclusive) way to handle some of this is by augmenting the data with transposition. It seems that the authors are not doing this here. Why not? It usually helps.

Another contribution appears to be the use of a per-time measure of loss. This is reasonable, and I believe others have done this as well. I certainly appreciated the explicit justification for it, however.

Note that the idea of using a vector to indicate metric subdivision was also used in (Johnson 2015).

Playing through some of the scores, it is clear that melodies themselves are often quite unusual (check user studies), but the voices do stay closely connected harmonically, which is what gives the system a certain aural coherence. I would be interested to hear (and look at) what is generated in two-part harmony, and even what is generated-- as a sort of baseline-- with just a single part.

I encourage the authors to look at and listen to the work by Johnson:
-listening samples: http://www.hexahedria.com/2015/08/03/composing-music-with-recurrent-neural-networks/
-associated publication: http://www.hexahedria.com/files/2017generatingpolyphonic.pdf

Overall, I think that the problem of generating rhythmically and polyphonically complex music is a good one, the approaches seem to generally be reasonable, although they do not appear to be particularly novel, and the musical results are not particularly impressive. The architectural choices are not always clearly presented.

---

> ### Author Response · Authors · 2018-11-23
> **re: many comments**
>
> Thank you for your extensive comments, and in particular for drawing our attention to the Johnson paper. Our relative pitch weight-sharing is the same idea as Johnson’s tied parallel networks, and we have made sure to recognize this in the new revision of the paper.
>
> We’ve made an effort to clean up many of your specific comments regarding clarity (see our new top-level post for a summary of changes to the paper) and we also hope that the new Appendix B gives a more holistic response to your requests for clarification. Specifically regarding the models in Table 5: all models in Table 5 address the single-part (homophonic) task, so there is no global model. All models in Table 3 follow the coupled (standard) run architectures described in Equations (2), (3), and (4).
>
> Regarding piano scores: we hope Appendix B provides some clarification, in particular subsection B.4. A piano score like the one in Figure 5 can be handle as two homophonic parts (bass staff and treble staff). The example given in Appendix A is a particularly complicated case where the lines split into 4-part polyphony; our point is that KernScores gives us labels to decompose these more complicated cases into homophonic parts that can be modeled using coupled architectures.
>
> Regarding the “unusualness” of the melodies, we point out that about half our corpus consists of Renaissance music. This may account for some of the difference in model output compared to models trained on corpora consisting entirely of music from the canonical classical era (17th-19th centuries).
>
> We have added single-part and two-part scores on the demos page, with the caveat that there are no single-part scores in our dataset and very few two-part scores, so these scores are somewhat out-of-sample.
>
> Regarding data augmentation: we found in our experiments that, for the relative pitch models, pitch-shifting data augmentation doesn’t improve log-loss. Likewise, given weight-sharing for parts, shuffling the order of the parts doesn’t improve log-loss.
>
> For the listening test, excerpts were chosen at random locations in the score. The fact that participants struggle to distinguish between training data and model outputs puts at least a lower bound on the quality of generated output. But we agree that listening to these excerpts without context can often make very little sense. On the other hand, we make no claim to model long-term dependencies in music, so presenting listeners with long clips doesn't elicit informative feedback. We are open to ideas about better ways to evaluate these models.

---

### Official Review · AnonReviewer3 · 2018-11-03
**Contains a good overview and extensive simulations. Unfortunately poor technical writing.**

**Rating:** 3
**Confidence:** 4

**Review:**


Composing polyphonic music is a hard computational problem.
This paper views the problem as modelling a probability distribution
over musical scores that is parametrized using convolutional and recurrent
networks. Emphasis is given to careful evaluation, both quantitatively and qualitatively. The technical parts are quite poorly written.

The introduction is quite well written and it is easy to follow. It provides a good review that is nicely balanced between older and recent literature.

Unfortunately, at the technical parts, the paper starts to suffer due to sloppy notation. The cross entropy definition is missing important details. What does S exactly denote? Are you referring to a binary piano roll or some abstract vector valued process? This leaves a lot of guess work to the reader.
Even the footnote makes it evident that the authors may have a different mental picture -- I would argue that a piano roll does not need two bits. Take a binary matrix: Roll(note=n, time=t) = 1 (=0) when note n is present (absent) at time t.

I also think the term factorization is sometimes used freely as a synonym for representation in last paragraphs of 4 and first two paragraphs of 5 -- I find this misleading without proper definitions.

The models, which are central to the message of the paper, are not described clearly. Please
define function a(\cdot) in (2), (3), (4), : this maybe possibly a typesetting issue (and a is highly likely a sigmoid) but what does x_p W_hp x x_pt etc stand for? Various contractions? You have only defined the tensor as x_tpn. Even there, the proposed encoding is difficult to follow -- using different names for different ranges of the same index (n and d) seems to be avoiding important details and calling for trouble. Why not just introduce an order 4 tensor and represent everything in the product space as every note must have a duration?

While the paper includes some interesting ideas about representation of relative pitch, the poor technical writing makes it not suitable to ICLR and hard to judge/interpret the extensive simulation results.

Minor:

For tensors, 'rank-3' is not correct use, please use order-3 here if you are referring to the number of dimensions of the multiway array.

What is a non-linear sampling scheme? Please be more precise.

The Allan-Williams citation and year is broken:
Moray Allan and Christopher K. I. Williams. Harmonising Chorales by Probabilistic Inference. Advances in Neural Information Processing Systems 17, 2005.

---

> ### Author Response · Authors · 2018-11-23
> **re: writing and piano roll representation**
>
> Thank you for your feedback. We have revised the paper to clarify the notational and definitional issues you identified.
>
> Regarding the number of bits needed for a piano roll: we have included further discussion of this point in Appendix B. We draw your attention in particular to Figures 5 and 7, which clarify the need for a second bit to distinguish between the onset of a note and continuation of a note from an earlier frame.

---

### Official Review · AnonReviewer1 · 2018-11-05
**well-written paper**

**Rating:** 7
**Confidence:** 3

**Review:**

The paper is well written and presented, giving a good literature review and clearly explaining the design decisions and trade-offs. The paper proposes a novel factorisation approach and uses recurrent networks.

The evaluation is both quantitative and qualitative. The qualitative experiment is interesting, but there is no information given about the level of musical training the participants had. You would expect very different results from music students compared to the general public. How did you control for musical ability/ understanding?

The paper has a refreshing honesty in its critical evaluation of the results, highlighting fundamental problems in this field.

Overall, while I am not an expert in musical composition and machine learning, the paper is clear, and appears to be advancing the art in a reliable fashion.

---

> ### Author Response · Authors · 2018-11-23
> **re: qualitative evaluation**
>
> Seven out of our twenty study participants self-identified as musically educated. Conditioned on that group, we found the following results:
>
> Clip Length: 10, 20, 30, 40, 50
> Average: 4.9, 6.0, 6.4, 6.9, 7.0
>
> So there was no significant distinction in results based on musical education.
>
> One thing to keep in mind is that a substantial fraction of our corpus is Renaissance music, which even well-educated classical musicians may be less familiar with. We informed participants of the scope of the training data prior to the listening test, but the bias towards Renaissance patterns in both the training data and model output could make classical music education less informative for discrimination.

---

### Author Response · Authors · 2018-10-29
**Demos**

We've sampled some scores from the model described in the paper and released them anonymously here:

http://ec2-18-219-197-207.us-east-2.compute.amazonaws.com/

Code for loading the KernScores dataset discussed in the paper will be made available once this submission is de-anonymized.

---

### Author Response · Authors · 2018-11-23
**revision summary**

We have updated the paper. This comment summarizes the substantial changes.

(*) We have included an additional Appendix B that precisely describes several popular factorizations of the distribution over scores, including the one used in this paper (Appendix B.5).

(*) Some clarifications of the cross-entropy metric in Section 3.

(*) Multiple revisions in Section 4 to clarify notation.

(*) Clarification of the distinction between a monophonic and homophonic prediction task in Section 4.

(*) Revision to Table 3: renaming “bottom” and “top” to “part” and “global” respectively, along with clarification in the caption regarding the meaning of these terms.

---

### Meta-Review · Area_Chair1 · 2018-12-14
**recurrent models for polyphonic music composition, quality seems to be the issue**

**Confidence:** 3
**Recommendation:** Reject

**Metareview:**

This paper proposes novel recurrent models for polyphonic music composition and demonstrates the approach with qualitative and quantitative evaluations as well as samples. The technical parts in the original write-up were not very clear, as noted by multiple reviewers. During the review period, the presentation was improved. Unfortunately the reviewer scores are mixed, and are on the lower side, mainly because of the lack of clarity and quality of the results.